# Glutamate Uptake Is Not Impaired by Hypoxia in a Culture Model of Human Fetal Neural Stem Cell-Derived Astrocytes

**DOI:** 10.3390/genes13030506

**Published:** 2022-03-12

**Authors:** Vadanya Shrivastava, Devanjan Dey, Chitra Mohinder Singh Singal, Paritosh Jaiswal, Ankit Singh, Jai Bhagwan Sharma, Parthaprasad Chattopadhyay, Nihar Ranjan Nayak, Jayanth Kumar Palanichamy, Subrata Sinha, Pankaj Seth, Sudip Sen

**Affiliations:** 1Department of Biochemistry, All India Institute of Medical Sciences, New Delhi 110029, India; vadanya.shrivastava@gmail.com (V.S.); devanjandey@gmail.com (D.D.); ankitsingh.skylark@gmail.com (A.S.); parthoaiims@hotmail.com (P.C.); jayanthaiims@gmail.com (J.K.P.); sub_sinha@hotmail.com (S.S.); 2Department of Molecular and Cellular Neuroscience, National Brain Research Centre, Manesar 122052, India; singalchitra@gmail.com (C.M.S.S.); paritoshjaiswal79@gmail.com (P.J.); pseth.nbrc@gov.in (P.S.); 3Department of Obstetrics and Gynaecology, All India Institute of Medical Sciences, New Delhi 110029, India; jbsharma2000@gmail.com; 4Department of Obstetrics and Gynecology, UMKC School of Medicine, Kansas City, MO 64108, USA

**Keywords:** hypoxic injury, fetal neural stem cells, astrocytes, excitotoxicity, glutamate uptake

## Abstract

Hypoxic ischemic injury to the fetal and neonatal brain is a leading cause of death and disability worldwide. Although animal and culture studies suggest that glutamate excitotoxicity is a primary contributor to neuronal death following hypoxia, the molecular mechanisms, and roles of various neural cells in the development of glutamate excitotoxicity in humans, is not fully understood. In this study, we developed a culture model of human fetal neural stem cell (FNSC)-derived astrocytes and examined their glutamate uptake in response to hypoxia. We isolated, established, and characterized cultures of FNSCs from aborted fetal brains and differentiated them into astrocytes, characterized by increased expression of the astrocyte markers glial fibrillary acidic protein (GFAP), excitatory amino acid transporter 1 (EAAT1) and EAAT2, and decreased expression of neural stem cell marker Nestin. Differentiated astrocytes were exposed to various oxygen concentrations mimicking normoxia (20% and 6%), moderate and severe hypoxia (2% and 0.2%, respectively). Interestingly, no change was observed in the expression of the glutamate transporter EAAT2 or glutamate uptake by astrocytes, even after exposure to severe hypoxia for 48 h. These results together suggest that human FNSC-derived astrocytes can maintain glutamate uptake after hypoxic injury and thus provide evidence for the possible neuroprotective role of astrocytes in hypoxic conditions.

## 1. Introduction

Hypoxic ischemic injury to the fetal and neonatal brain is one of the most common causes of various neurological disabilities, and even death in children worldwide [1]. Several pregnancy complications, such as maternal intrauterine infections, preterm birth, preeclampsia, and maternal and fetal systemic inflammation can cause hypoxic ischemic injury to the fetal brain [2,3]. Currently, there is no treatment for survivors of such injuries who suffer lifelong ailments from neurological sequelae. Numerous studies suggest differences between immature and mature brain in the pathophysiology of the consequences of brain injury [4,5]. Hypoxic ischemia is also known to damage selected regions of the immature brain at different ages [6], and therefore, it is critically important to understand the impact of hypoxic ischemic injury on various cell types of the fetal/neonatal brain. 

The central nervous system comprises of different cell types, including neurons and non-neuronal glial cells. Astrocytes, conventionally considered as supporting glial cells of the brain, also account for other important roles, such as providing energy intermediates to neurons, maintaining the water and ionic balance in and around them, the formation and regulation of the blood-brain barrier, and calcium signaling, along with release and uptake of neurotransmitters, especially that of glutamate and γ-amino butyric acid (GABA) [7]. Glutamate is an important excitatory neurotransmitter in the brain and astrocytes play a key role in preventing glutamate excitotoxicity by taking up excess glutamate from the synapse via the excitatory amino acid transporters (EAATs). 

Based on the studies in experimental models, glutamate excitotoxicity is believed to be one of the mechanisms involved in the neuronal damage associated with hypoxic brain injuries [8]. While most of these studies conducted in rodent models provide important insights into the mechanisms of hypoxic ischemic injuries in mammalian brains, fundamental differences exist between the rodent and human brain, which differ not only in size and number of neurons, but also in morphological and functional diversity, as well as gene expression and its regulation [9]. Nonetheless, the mechanism of hypoxic brain injury is not fully understood, and further studies are particularly warranted to investigate the roles and contributions of various human fetal/neonatal neural cells in the development of glutamate excitotoxicity and in hypoxic ischemic brain injuries.

This study aimed to evaluate the role of astrocytes in hypoxic injury by developing an in vitro model, comprising of human astrocytes derived from fetal neural stem cells. Using this model system, the effect of hypoxia on astrocyte function was evaluated, particularly the expression and function of glutamate transporters.

## 2. Results

### 2.1. Isolation and Characterization of Human Fetal Neural Stem Cells

Neural stem cells were isolated from the subventricular zone of the fetal brain (n = 5). Neurospheres were observed after 48–72 h in culture which showed fetal neural stem cells (FNSCs) radiating outward from the core (Figure 1A). Neurospheres were dissociated and sub-cultured, to form monolayers. FNSCs cultured in monolayer appeared as small cells with unipolar morphology (Figure 1B). Human FNSCs were expanded and characterized by the expression of Nestin on immunocytochemical staining (Figure 1C). 

### 2.2. Differentiation of Human FNSCs into Astrocytes

After 3–4 passages, FNSCs were differentiated into astrocytes (n = 5) by replacing the neural stem cell media with Complete Minimum Essential Media (CMEM), and differentiation was monitored over 14 days. With the progress of differentiation, cells were observed to change morphology from small unipolar cells to large, flat cells with large nuclei (Figure 1D,E). Cells were harvested at days 0, 7, and 14 to evaluate change in expression of specific markers. 

### 2.3. Expression of Astrocytic Markers in Differentiating Cells

There was an increase in the mRNA expression of the astrocyte-specific marker glial fibrillary acidic protein (GFAP) from day 0 to day 7 (fold change 4.59 ± 2.48, *p*-value > 0.05) and day 14 (fold change 8.6 ± 5, *p* = 0.0079) (Figure 2A). These changes at the mRNA level were mirrored in GFAP protein expression by Western blotting (Figure 2C). An increase was observed in the normalized expression of GFAP from 0.28 ± 0.17 at day 0, to 1.03 ± 0.2 at day 7 (*p* > 0.05), and 1.55 ± 0.05 at day 14 (*p* = 0.0134), equating to a fold change of 7.2 ± 4.4 at day 14 (Figure 2D). Differentiating cells also showed an increase in the mRNA expression of astrocyte specific glutamate transporters (Figure 2B). SLC1A3 (EAAT1) mRNA expression increased with a fold change of 1.4 ± 0.3 (*p* > 0.05) at day 7 and fold change of 2.90 ± 0.67 at day 14 (*p* = 0.02). SLC1A2 (EAAT2) mRNA also increased over the course of differentiation, with fold changes of 1.1 ± 0.08 (*p* > 0.05) at day 7 and fold change of 2.93 ± 1.9 (*p* = 0.019) at day 14 of differentiation. Differentiating cells (day 0 and day 14) were also analyzed for the expression of GFAP, by flow cytometry (Figure 3A–D). The percentage of cells expressing GFAP increased significantly from 36 ± 7.8 percent (day 0) to 94.97 ± 1.7 percent (day 14) (*p* = 0.03) (Figure 3E). 

### 2.4. Expression of NSC-Marker Nestin and Neuronal Marker Doublecortin by Differentiating Cells

Differentiating cells were also evaluated for the expression of the neural stem cell marker, Nestin, at different time points during differentiation, using quantitative PCR. Nestin (NES) mRNA expression was found to decrease as differentiation progressed, with a fold change of 0.37 ± 0.25 at day 14 (*p* = 0.18) (Figure 2E). A significant difference was observed among all the groups (*p* = 0.0432, Kruskal-Wallis test), although significance was not achieved between any two individual groups. mRNA expression of Doublecortin (DCX) (Figure 2F), a microtubule-associated protein expressed during neuronal differentiation, decreased from day 0 to day 7 (fold change 0.28 ± 0.26, *p* > 0.05) and subsequently to day 14 (fold change 0.14 ± 0.21, *p* = 0.03), thereby confirming absence of neurogenesis during the differentiation protocol. 

From these observations, it was evident that human FNSCs successfully differentiated into astrocytes and based on the expression pattern of cell type specific markers, cells at day 14 were subjected to further hypoxia experiments. 

### 2.5. Exposing Differentiated Astrocytes to Hypoxia

Differentiated astrocytes at day 14 with 70–80% confluency were subjected to oxygen concentrations mimicking normoxia (20% and 6%), moderate hypoxia (2%) and severe hypoxia (0.2%) for 48 h. After exposure to hypoxia, morphological changes did not indicate patterns associated with cell death. Exposure to hypoxia was validated by studying the expression of Hypoxia inducible factor 1α (HIF1α). Western blot analysis showed an increase in the expression of HIF1α protein with increasing grade of hypoxia (Figure 4A). Carbonic anhydrase 9 (CA9), vascular endothelial growth factor (VEGF) and phosphoglycerate kinase (PGK1) are markers known to be upregulated in hypoxia. Gene expression analysis of CA9 observed in various grades of hypoxia (Figure 4B) showed a significant increase in moderate (fold change 10.8 ± 10.9, *p* = 0.04) and severe hypoxia (fold change 55.99 ± 53, *p* < 0.0001) as compared to normoxia (20% oxygen). A slight increase in CA9 expression was seen in cells exposed to 6% oxygen (fold change 2.1 ± 1.8) but was not found to be statistically significant. VEGF and PGK1 gene expression also increased with hypoxia. Samples that showed upregulation of these hypoxia-responsive genes, were analyzed further, along with normoxic controls.

### 2.6. Evaluation of Glutamate Transporter Expression in Astrocytes Exposed to Hypoxia

Expression of the major glial glutamate transporter EAAT2 in astrocytes exposed to hypoxia was studied using qPCR (Figure 4C) and Western blot (Figure 4D,E). Gene expression analysis of astrocytic SLC1A2 (EAAT2) at different oxygen concentrations showed log_2_ fold change (compared to 20% oxygen) values of 0.76 ± 1.4 at 6% oxygen, 0.68 ± 2 at 2% oxygen and 0.7 ± 1.4 at 0.2% oxygen. Statistical analysis showed no significant difference between groups. Western blot analysis showed astrocytic expression of EAAT2 (normalised to β-actin) of 0.94 ± 0.14 at 20% oxygen, 1.05 ± 0.05 at 6% oxygen, 1.02 ± 0.02 at 2% oxygen and 1.14 ± 0.07 at 0.2% oxygen. There was no significant difference detected between groups.

### 2.7. Evaluation of Glutamate Uptake in Astrocytes Exposed to Hypoxia

To further delineate the effects of hypoxia on astrocytic glutamate transport, a glutamate uptake assay was performed on astrocytes exposed to hypoxia. The glutamate concentrations in the supernatant decreased after 30 mins incubation with 2 mM glutamate, in all treatment groups including normoxic controls (Appendix A), indicating that glutamate was predominantly being taken up by the cells. Normoxic astrocytes showed a glutamate uptake of 4.4 ± 2.5 micromoles/mg of protein at 20% oxygen and 4.7 ± 1.9 micromoles/mg of protein at 6 % oxygen. Astrocytes maintained glutamate transport at 2% oxygen (glutamate uptake 3.5 ± 2.2 μmoles/mg of protein) and 0.2% oxygen (glutamate uptake 5.1 ± 2.8 μmoles/mg of protein) (Figure 4F). Statistical analysis showed no significant difference between groups. 

## 3. Discussion

Human fetal neural stem cells were isolated from the subventricular region of the brain tissue of aborted fetuses (12–20 weeks gestation) obtained after medical termination of pregnancy. FNSCs showed characteristic features like formation of neuro-spheres in culture, and expression of the neural stem cell marker, Nestin. 

Following the induction of differentiation in FNSCs, a change in morphology of the cells from small unipolar phenotype of FNSCs, to large, flat polygonal morphology of astrocytes, was observed as differentiation progressed. Although this morphology differs from the classical astrocyte morphology demonstrated in vivo, ample evidence exists that in vitro cultures that use serum as a differentiation-inducing agent, display a fibroblast-like astroglial appearance, similar to that observed in our study [10]. Concomitantly, a gradual rise in GFAP expression peaking at day 14 of differentiation, corresponding with decreasing Nestin expression was observed throughout the course of differentiation. GFAP is the canonical astrocytic marker and has been used for the characterization of astrocytes in numerous studies [7]. A decrease in the expression of Nestin observed by us indicates decrease in neural stem cell population as they differentiate into astrocytes. Other studies have observed a similar decline in expression of Nestin following differentiation of neural progenitors into neuronal or glial lineage [11]. 

Flow cytometric analysis showed that nearly 95% of cells were expressing GFAP at day 14 of differentiation. Interestingly, 36% of fetal neural stem cells also expressed GFAP. This finding corroborates a previous study, which showed that progenitor cells in adult/embryonic tissue express GFAP [12]. 

In our study, the differentiating cells at day 14 not only expressed GFAP, but also expressed the astrocyte specific glutamate transporter EAAT1 and EAAT2, as well as took up glutamate from the extracellular media, indicating functional activity. 

On the basis of the observation that peak GFAP expression was detected at day 14 of differentiation, astrocytes were exposed to different oxygen concentrations at this time-point. Even though cells are usually maintained at 20% oxygen concentration during routine cell culture, various studies reported physiological oxygen concentration in the brain to be much lower, ranging from 1% to 6% [13]. Therefore, we exposed astrocytes to oxygen concentrations mimicking normoxia and conditions that may prevail under normoxia in vivo (20% and 6%), moderate hypoxia (2%), and severe hypoxia (0.2%) for 48 h. Cellular response to hypoxia involves the induction of the Hypoxia Inducible Factor 1 α (HIF1α) pathway. In our study we saw an increase in the protein expression of HIF1α with an increase in the grade of hypoxia. HIF1α also stimulates the expression of pro-survival and pro-angiogenic molecules such as vascular endothelial growth factor (VEGF), phosphoglycerate kinase (PGK-1) and carbonic anhydrase (CA9), all of which have been demonstrated to be raised following hypoxia [14] and are relatively stable. In our study, we observed that CA9 was robustly increased following exposure of differentiated astrocytes to hypoxia, resulting in a 40–60-fold increase in its expression, consistent with previous studies of hypoxia exposure in astrocytes [15]. This was further validated by observing a gradient increase in VEGF and PGK-1 expression with increasing hypoxia. 

The preterm infant is more susceptible to hypoxic-ischemic brain injury as a consequence of underdeveloped vasculature, vulnerable glial populations and immature vasoregulatory reflexes [16,17]. Oxygen deprivation or hypoxia precipitated as a consequence of loss of cerebral autoregulation, is a key component in the pathophysiology of such injury [18]. An important contributor to neuronal death following hypoxic injury is glutamate excitotoxicity, where excessive stimulation of glutamate receptors can lead to neuronal dysfunction and death [19]. Released glutamate is negligibly processed extracellularly, thus its clearance depends on diffusion and uptake [20], mostly by astrocytes [21] and to a smaller extent, by neurons and oligodendrocytes [22]. Astrocytes, therefore, play a key role in preventing glutamate excitotoxicity by taking up excess glutamate from the synapse via the excitatory amino acid transporters (EAATs) out of which EAAT2 is astrocyte-specific and responsible for almost 90% of the glutamate uptake in the forebrain [23]. Interestingly, we observed that hypoxia exposure did not result in any significant change in the expression of the glutamate transporter EAAT2. Differentiated astrocytes in normoxia showed an uptake of glutamate from the extracellular space, which did not change on exposure to hypoxia. This corroborates previous reports that have shown that astrocytes are able to maintain viability in hypoxia, provided energy substrates are present [24]. Astrocytic EAAT2 was found to be upregulated in rat models of chronic brain ischemia as well as human tissue [25] and astrocytes have been shown to continue serving neuroprotective roles in models of intense oxidative stress [26]. It has previously shown that hypoxia can act synergistically with inflammatory mediators released by astrocytes to stimulate glutamate release [27]. However, in contrast to this study which used a 24 h incubation with IL-B to measure glutamate release, our 30 min incubation for measuring uptake might not be sufficient to cause significant accumulation of inflammatory mediators, preventing such factors from influencing measurement of glutamate uptake.

Rodent models of neurological diseases have provided valuable insights in the pathogenesis of neurodevelopmental diseases. However, important differences exist between rodent and human neurological brain models due to the higher complexity of the human brain as well as its differing metabolic and transcriptomic programs [28]. Most studies on astroglial glutamate uptake have used rodent models and studies like ours, on primary cultures of non-immortalized human astrocytes, are scarce. To the best of our knowledge, ours is the first study that demonstrates the effects of hypoxia exposure on glutamate uptake in human astrocytes differentiated from fetal neural stem cells. The biological differences between human and rat models of such diseases could be responsible for the differing results between our study and that of Dallas et al. [29].

Even though our findings corroborate studies asserting that astrocytes are resistant to hypoxic conditions [30,31,32], they are in contrast to studies where hypoxic-ischemic injuries to the brain lead to a reduction in glutamate transporter expression [33]. This discrepancy may be due to the added insult of substrate deprivation seen in these studies. Indeed, hypoxia and ischemia have been shown to have differing effects on astrocytes with previous evidence also suggesting that astrocytes are able to maintain viability and function during hypoxia if glucose is present [32,34].

Some limitations of our study also need to be considered. The time duration of 48 h of hypoxia exposure that we have used for our experiments is only a representative selection based on preliminary work done in our lab and other similar studies [35,36]. However, there is no designated timepoint for carrying out such experiments as different conditions of hypoxic injury may have different exposure time points. Although astrocytes are also known to release glutamate in response to stimuli such as intracellular calcium changes [37], our study does not focus on this aspect of calcium flux on astroglial function. Also, other features of ischemic injury such as neuroinflammation, may influence the neuroprotective effect of astrocytes in hypoxic injury, and these mechanisms may need to be further validated in the future. Further, neuron-astrocyte cocultures can provide valuable information regarding contribution from neurons, as may be relevant in vivo. Therefore, our findings merit further investigations using co-culture or in vivo systems. 

Our study is novel as we have developed a homogenous population of primary human astrocytes from human FNSCs isolated from aborted fetal brain tissue. The astrocytes thus derived, were characterized with astrocyte-specific markers. Differentiated astrocytes were functional, as indicated by the expression of glutamate transporters and by the uptake of extracellular glutamate by these cells. Moreover, it was observed that these differentiated astrocytes maintained glutamate transporter expression and function, even after exposure to moderate and severe hypoxia. Our unique in vitro model of human FNSC derived astrocytes exposed to hypoxic injury, not only corroborates the existing evidence supporting the relative resistance of astrocytes to hypoxic injury, but further substantiates it by demonstrating that astrocytes exposed to hypoxia, maintain glutamate uptake. These results also give an indirect proof of the neuroprotective role of astrocytes in hypoxic brain injury, including conditions of severe hypoxia. 

## 4. Methodology

### 4.1. Sample Collection

Aborted fetal samples were collected (n = 5), after taking informed consent from mothers undergoing Medical Termination of Pregnancy (MTP) in the Department of Obstetrics and Gynecology, AIIMS, New Delhi, India. Necessary approvals were taken from Institutional Ethics Committee and Institutional Committee for Stem Cell Research, before starting the study and the study was carried out in conformation with the World Medical Association Declaration of Helsinki. Fetal samples from mothers undergoing MTP in the second trimester of pregnancy (12–20 weeks) for maternal indications, were included, while fetal indications (such as chromosomal anomalies) were excluded from the study. 

### 4.2. Isolation of Human Fetal Neural Stem Cells (FNSCs)

Isolation of human fetal neural stem cells (FNSCs) from aborted fetuses was done in accordance with previous protocols [38]. Briefly, FNSCs were isolated from the subventricular zone of the fetal brain and subsequently plated onto poly-d-lysine coated culture flasks in neural stem cell media containing Neurobasal media (Catalog #21103049, GIBCO, Waltham, MA, USA) with 1% N2 supplement (Catalog #17502048, GIBCO, NY, USA), 2% Neural survival factor-1 (Catalog #CC-4323, Lonza, Bend, OR, USA), 1% Glutamax (Catalog #35050061, GIBCO, NY, USA), 5mg/mL of bovine serum albumin (BSA) (Catalog #A9418, Sigma, St. Louis, MO, USA), penicillin (50 IU/mL), streptomycin (50 µg/mL) (Catalog #15070-063), GIBCO, NY, USA and gentamicin (2 µg/mL). Neuro-spheres were observed after 2–3 days, dissociated by gentle agitation during sub-culture, and then seeded onto poly-d-lysine coated flasks to allow their adherence, to generate monolayers of FNSCs. Differentiation into astrocytes was initiated after 2–3 passages. 

### 4.3. Differentiation of Human Fetal Neural Stem Cells into Astrocytes

FNSC growth media was substituted by Complete Minimal Essential Media (CMEM) consisting of Minimal essential media (MEM) (Catalog # M0268, Sigma, MO, USA) with 10% fetal bovine serum (FBS Catalog #RM112, Himedia, Mumbai, MH, INDIA) to induce differentiation of FNSCs into astrocytes [38]. Half the media was changed every alternate day for 14 days, and sub-culturing was done when cells reached 80% confluency. Astrocytes were characterized by presence of glial fibrillary acidic protein (GFAP) and excitatory amino acid transporters (EAAT1 and EAAT2) in cells. 

### 4.4. Exposure of Differentiated Human Astrocytes to Different Oxygen Concentrations

Differentiated human astrocytes (at day14), at 70–80% confluency, were exposed to oxygen concentrations mimicking normoxia (20% and 6%) and hypoxia (2% and 0.2%) for 48 h, at 37 °C and 5% CO_2_ that was created using an Anoxomat hypoxia induction system (Advanced Instruments, Norwood, MA, USA). Hypoxia exposure was validated by evaluating the expression of HIF1α, CA9, VEGF and PGK-1.

### 4.5. RNA Isolation, cDNA Synthesis and qPCR

Total RNA was extracted from the cells at different stages of differentiation after exposure to various oxygen concentrations, using Tri-Reagent (TR118, Sigma, St. Louis, MO, USA) and quantified by Nano-Drop ND-1000 spectrophotometer (Thermo-Fisher Scientific, Waltham, MA, USA). cDNA was synthesized with 1 µg total RNA using M-MuLV-RT (Catalog #EP0442, Thermo-Fisher Scientific, MA, USA) and random hexamer primers (Catalog #30142, Thermo-Fisher Scientific, MA, USA) as described previously [39]. The expression of various genes was evaluated in the cells (in triplicates) using gene-specific primers (IDT, IL, USA) (Table 1) and DyNAmo Flash SYBR Green qPCR kit (Catalog # F416L, Thermo-Fisher Scientific, MA, USA) using CFX96 Touch™ Real-Time PCR Detection System (BioRad, Hercules, CA, USA). 18S rRNA was used as an internal reference gene for normalization. Relative fold change in gene expression was calculated using 2^−∆∆CT^ method. For differentiation experiments, day 0 samples (human FNSCs) were used as controls, while differentiated astrocytes (day 14) exposed to 20% oxygen (mimicking normoxia) were the controls for hypoxia experiments. 

### 4.6. Western Blotting

Protein was isolated using Tri-Reagent (TR118, Sigma, MO, USA) as per the manufacturer’s protocol and quantified by bicinchoninic acid (BCA) method (Catalog # 23225, Thermo-Fisher Scientific, MA, USA). Equal amounts of protein extracts (20 µg) were electrophoresed on 10–15% SDS-polyacrylamide gels and electro-transferred onto nitrocellulose membrane (BioRad, Hercules, CA, USA). The membrane was blocked with 5% non-fat milk (NFM) dissolved in 0.1% Tween-20 containing tris-buffered saline containing (TTBS) for 1 h. The blots were probed overnight with specific primary antibodies (Table 2), which were diluted in 1% NFM in TTBS. After washing with TTBS, the membrane was incubated for 1 h with the appropriate secondary antibody (Table 2). The blot was incubated with Luminol and peroxidase (Abbkine SuperLumia ECL Plus Kit Catalog # K22030, Wuhan, China) and chemiluminescence detection was done using Azure Biosystems c280 gel documentation sytem (Dublin, CA, USA), followed by analysis with Image J software. Normalization was done using glyceraldehyde 3 phosphate dehydrogenase (GAPDH) and β-actin protein levels.

### 4.7. Flow Cytometry

Cells (at different stages of differentiation) were fixed using 2% paraformaldehyde (PFA) and then permeabilized with 1% BSA containing 0.1% Triton X-100. Cells were then blocked in 2% BSA for half an hour and subsequently stained (intracytoplasmic) with Alexa Fluor 647 conjugated rabbit anti-human GFAP antibody (BD Biosciences, cat. no. 561470) using appropriate controls. Cells were washed, resuspended in 2% paraformaldehyde, and data was acquired using BD LSR Fortessa (BD Biosciences, San Jose, CA, USA) and analyzed using FlowJo v10 software. 

### 4.8. Immunocytochemistry

Cells were plated onto coverslips, washed once with PBS and fixed with 4% PFA. The cells were then incubated for 1 h in 1% BSA with 0.5% Triton X-100 and then washed with PBS before incubating overnight at 4 °C with primary antibody (Rabbit Anti-Nestin 1:1000 (Millipore, Burlington, MA, USA, Cat no. ABD69). Thereafter, cells were washed thrice with PBS and then incubated for an hour in secondary antibody (Mouse anti-Rabbit FITC 1:1000 (Invitrogen, Waltham, MA, USA, Cat no. A11008) for 1 h at room temperature. Cells were then washed thrice with PBS, and mounted onto glass slides using Vectashield (Catalog # H1200, Vector Labs, Burlingame, CA, USA) mountant containing DAPI. The slide was allowed to dry overnight. Images were taken on Nikon Eclipse Ti-S fluorescent microscope (Tokyo, Japan) and analyzed with NIS-Elements BR 4.30.00 64-bit software. 

### 4.9. Glutamate Uptake Assay

Glutamate uptake by astrocytes was measured as described previously [40] with minor modifications. Briefly, after exposing astrocytes to hypoxia for 48 h, supernatant in culture flasks were discarded and replaced with 2 mL of Hank’s Balanced Salt Solution (HBSS) (Catalog # 14175095, ThermoFisher, Waltham, MA, USA) with 2 mM monosodium glutamate (Catalog #G5889, Sigma, MO, USA). Cells were then incubated for 30 min at 37 °C in normoxic conditions (20% O_2_), following which, the supernatant was removed and snap-frozen for estimation of glutamate, while the cells were treated with TRIzol for isolation of RNA and protein. Glutamate was estimated in the supernatant by a fluorimetric assay (Abcam cat.no. ab138883) which used a glutamate dehydrogenase coupled mechanism for estimation. Glutamate concentrations in 2 mL HBSS containing 2 mM glutamate (vehicle control) were estimated and found to be ~2.06 mM. Glutamate uptake values were normalized to the amount of protein obtained in the corresponding cells (n = 7).

### 4.10. Statistical Analysis

Statistical analysis was done using Graph Pad Prism v6. Kruskal Wallis test was used to detect significant differences between groups. In datasets that showed a significant *p*-value on the Kruskal-Wallis test, a comparison between specific groups was made by Dunn’s pairwise comparison test. *p*-value < 0.05 was considered statistically significant.

## Figures and Tables

**Figure 1 genes-13-00506-f001:**
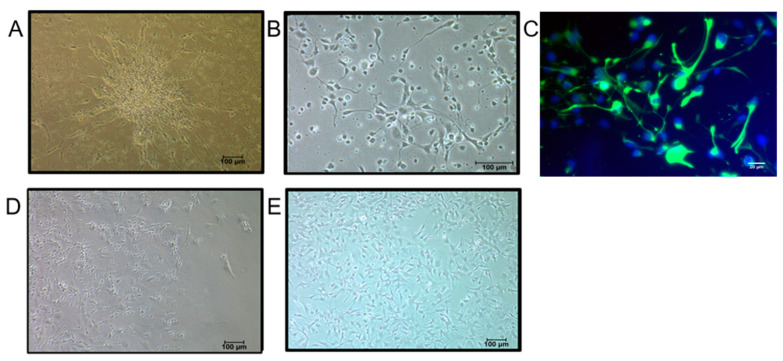
Morphological features of human fetal neural stem cells and differentiating astrocytes. (**A**) Neurosphere formation seen after 24–48 h of fetal neural stem cell (FNSC) isolation. (**B**) FNSCs in monolayer culture display unipolar morphology. (**C**) Immunofluorescence image of FNSCs stained for neural stem cell marker Nestin (green). Nuclei are stained with DAPI (blue). Morphological changes seen at (**D**) day 7, (**E**) day 14 of astrocytic differentiation. Differentiated cells are larger and show flattened morphology.

**Figure 2 genes-13-00506-f002:**
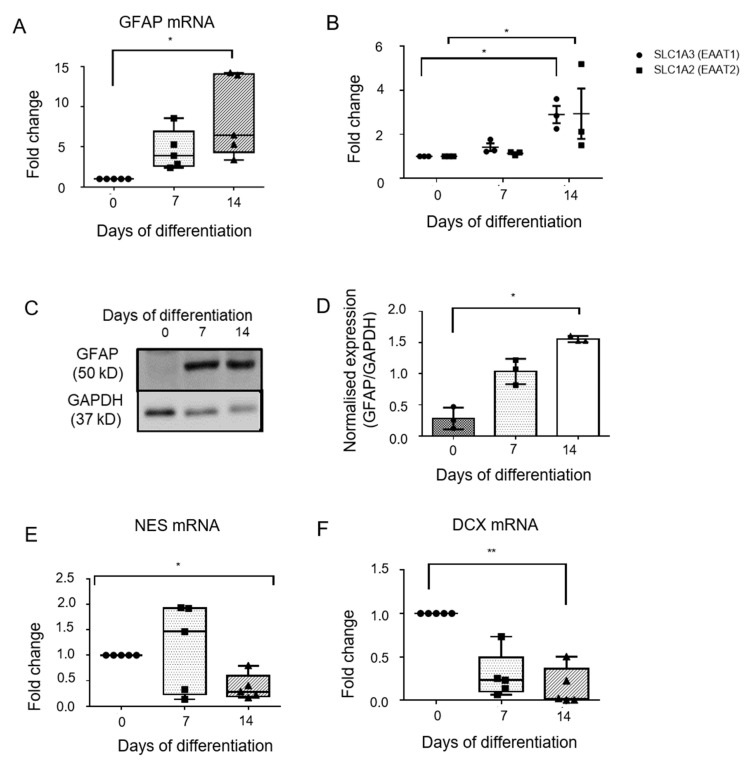
Expression of lineage markers during differentiation of human FNSCs into astrocytes. Expression of astrocytic marker (**A**) glial fibrillary acidic protein (GFAP) (n = 5), (**B**) Excitatory amino acid transporters SLC1A3 (EAAT1) and SLC1A2 (EAAT2) (n = 3) mRNA expression at various days of differentiation as obtained on qPCR analysis. (**C**) Representative western blot image showing protein expression of GFAP at various days of differentiation. Glyceraldehyde 3-phosphate dehydrogenase (GAPDH) is loading control. (**D**) Quantification of western blot data (n = 3). mRNA expression of neural stem cell marker NES (Nestin) (**E**) and neuronal marker DCX (Doublecortin) (**F**) at various days of differentiation as obtained on qPCR analysis (n = 5). qPCR data is represented as fold change in gene expression obtained using day 0 samples as control. Data represented as Mean ± SD. * *p*-value < 0.05, ** *p*-value < 0.01.

**Figure 3 genes-13-00506-f003:**
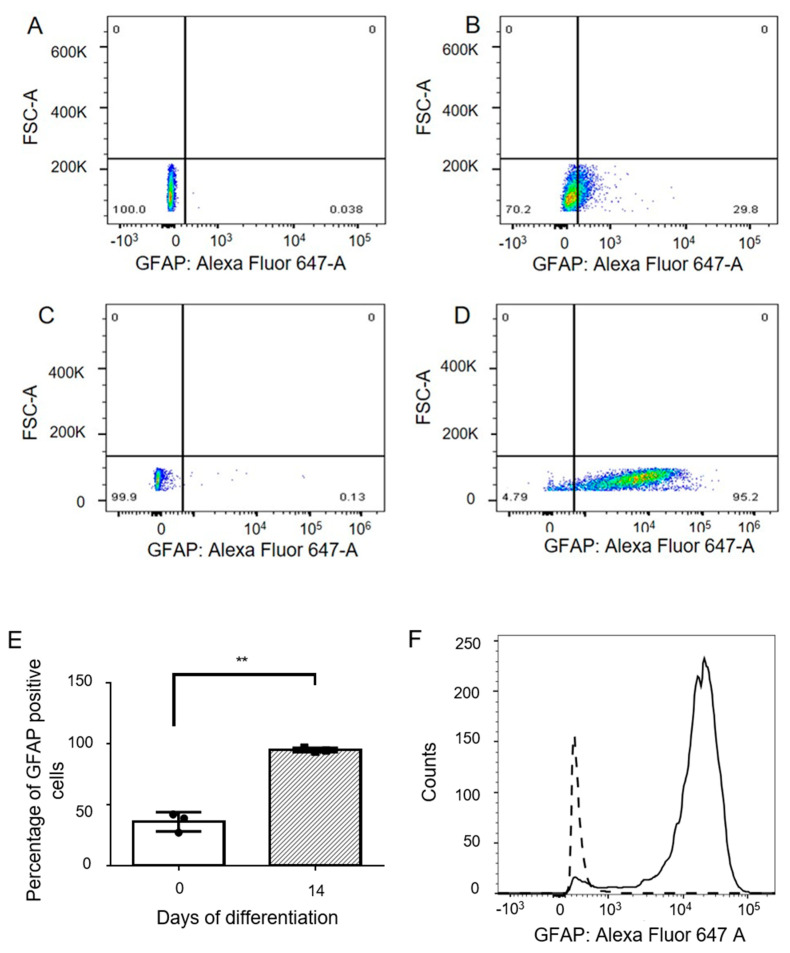
Flow cytometric analysis of GFAP expression during astrocytic differentiation. Dot plots showing (**A**) Unstained and (**B**) stained populations of human FNSCs (day 0). Representative dot plots for (**C**)unstained and (**D**) stained populations of cells at day 14 of astrocytic differentiation. (**E**) Quantification of flow cytometric analysis of GFAP expression in differentiating cells (n = 3). (**F**) Representative histogram showing increase in GFAP expression at day 14 compared to day 0 of differentiation. Data represented as Mean ± SD. ** *p*-value < 0.01.

**Figure 4 genes-13-00506-f004:**
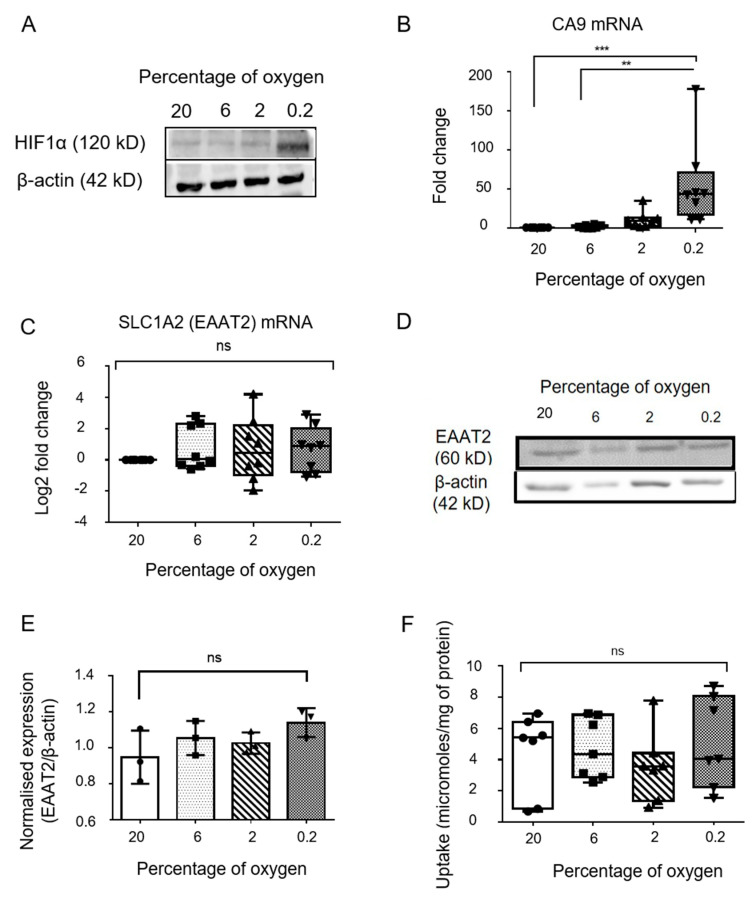
Hypoxia treatment to differentiated astrocytes. (**A**) Western blot image for Hypoxia-inducible factor 1α (HIF1α) expression in astrocytes exposed to various grades of hypoxia. (**B**) Carbonic anhydrase 9 (CA9) and (**C**) SLC1A2 (EAAT2) gene expression in astrocytes exposed to various grades of hypoxia as analysed by qPCR (n = 8). (**D**) Representative western blot picture and (**E**) quantification of western blot data showing expression of EAAT2 in various grades of hypoxia (n = 3). (**F**) Box and whisker plot showing glutamate uptake by astrocytes exposed to different concentrations of oxygen (n = 7). Data represented as Mean ± SD. ** *p*-value < 0.01, *** *p*-value < 0.001, ns: non-significant.

**Table 1 genes-13-00506-t001:** Sequences of primers used for gene expression analysis (FP = Forward primer; RP = Reverse primer).

Gene	Primer Sequence (5′ to 3′)
18s rRNA	FP: GTAACCCGTTGAACCCCATT RP: CCATCCAATCGGTAGTAGCG
GFAP	FP: AGCCCACTCCTTCATAAAGCC RP: ATGCGTCTCCTCTCCATCCT
Nestin	FP: CCTCAAGATGTCCCTCAGCC RP: TCCAGCTTGGGGTCCTGAAA
Doublecortin	FP: GGGGGTGTGGGCATAAAGAA RP: CCTGCTCTTTACCAGCCTCC
EAAT1	FP: GAATGGCGGCGCTAGATAGT RP: CCAGGCTTCTACCAGATTTG
EAAT2	FP: CAGGGAAAGCAACTCTAATC RP: CAAGGTTCTTCCTCAACA
CA9	FP: CTTTGAATGGGCGAGTGATT RP: CTTCTGTGCTGCCTTCTCATCT
VEGF	FP: ACCATGAACTTTCTGCTGTCTTG RP: ATGGCTTGAAGATGTACTCGATCTC
PGK-1	FP: CCGAGCCAGCCAAAATAGA RP: ACTTTAGCTCCGCCCAGGAT

**Table 2 genes-13-00506-t002:** Specifications for primary and secondary antibodies used for western blot experiments.

Antibody	Molecular Weight	Antibody Specifications	Antibody Dilution	Procured From
Anti-GFAP (Catalog #ABP54511)	50 kDa	Polyclonal Rabbit IgG	1:1000	Abbkine, Wuhan, China
Anti-EAAT2 (Catalog #ABP57320)	60 kDa	Polyclonal Rabbit IgG	1:1000	Abbkine, Wuhan, China
Anti-HIF1α (Catalog #36169S)	120 kDa	Monoclonal Rabbit IgG	1:1000	CST, Danvers, MA, USA
Anti-GAPDH (loading control for differentiation experiments) Catalog #10-10011	37 kDa	Monoclonal Mouse IgG	1:2000	Abgenex, Bhubaneswar, India
Anti β-actin (loading control for hypoxia experiments) Catalog #STJ94020	42 kDa	Polyclonal Rabbit IgG	1:1000	St. John’s Laboratory, London, UK
HRP-tagged Anti-rabbit secondary antibody (Catalog #7074S)	-	Goat Anti-rabbit IgG	1:2000	CST, Danvers, MA, USA
HRP-tagged anti-mouse secondary antibody (Catalog #7076S)	-	Horse Anti-mouse IgG	1:2000	CST, Danvers, MA, USA

## Data Availability

Authors are willing to provide the relevant experimental data upon request.

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
