# Peer review of "Glutamate Uptake Is Not Impaired by Hypoxia in a Culture Model of Human Fetal Neural Stem Cell-Derived Astrocytes"

_genes, 2022, doi:10.3390/genes13030506_

Round 1

Reviewer 1 Report

To Authors:

In this report, the authors established and characterized human fetal neural stem cells. Astrocytes derived from human fetal neural stem cells were utilized to examine the effects of hypoxia on glutamate uptake by astrocytes. They found that hypoxia did not affect the ability of glutamate uptake in human neural stem cell-derived astrocytes. This is an interesting observation in the field of neuroscience. However, it contains several concerns as listed below:

  • In Fig. 1B and C, the scale bars are not evident.
  • In Fig. 2B, the bar graphs should be overlaid with scatter plots like the other graphs.
  • It is not clear what the authors mean when they state, “Neuronal differentiation was ruled out by evaluating expression of neuronal marker Doublecortin.”
  • Please use official gene names and put them above graphs when the authors show results of mRNA expression levels.
  • In Fig. 4D and E, the labeling of X-axis and Y-axis must be incorrect, respectively.
  • Please explain the rationale for choosing 48 h of hypoxia exposure in more detail.
  • The authors are recommended to examine the effects of hypoxia on EAAT1 protein levels in astrocytes.
  • It has been demonstrated that hypoxia induces the release of interleukin-1beta (IL-1b) from astrocytes via HIF-1 (Zhang et al., J Neuroimmunol, 2006). In addition, it has been suggested that hypoxia and IL-1b cooperatively induce the release of glutamate from astrocytes (Fogal el al., J Neurosci, 2007; Vangeison and Rempe, Neuroscientist, 2009). Although glutamate uptake has been assessed using radiolabeled glutamate in many previous reports, the authors assessed glutamate uptake by measuring glutamate concentrations in culture supernatant. Therefore, it is possible that glutamate concentrations in HBSS were affected not only by glutamate uptake but also by glutamate released by astrocytes in the culture and assay system of this report. The authors are recommended to confirm if glutamate release from astrocytes is stimulated in their culture system.
  • The authors cited a report by Santello et al. (ref. 34) for Glutamate uptake assay. However, this is a review article and there is no information about the procedure of glutamate uptake assay in the report. Please cite an appropriate report.
  • Not all product number information is provided.

Author Response

Response to Reviewer's Comments

Manuscript ID: genes-1598699

Title: Glutamate uptake is not impaired by hypoxia in a culture model of human fetal neural stem cell-derived astrocytes.

We are thankful to the reviewers for critically reviewing the manuscript and giving their valuable suggestions for its improvement.  We are gratified that the reviewers find our work of importance to the field. All suggestions have been incorporated into the revised manuscript. We have addressed all points, including comments, questions, and suggestions given by reviewers and marked the major changes in the manuscript.

Reviewer #1

  1. In Fig. 1B and C, the scale bars are not evident.

Response: We have replaced Fig. 1B with another image having a clearer scale bar. In Fig. 1C, we have made the scale bar (20 µm) bold to be clearly visible.

  1. In Fig. 2B, the bar graphs should be overlaid with scatter plots like the other graphs.

Response: As suggested by the reviewer, we have modified Fig. 2B (similar to other graphs in Fig. 2).

  1. It is not clear what the authors mean when they state, “Neuronal differentiation was ruled out by evaluating expression of neuronal marker Doublecortin.”

Response: We are sorry for the confusion. Doublecortin is a microtubule-associated protein that is expressed during the differentiation of neural stem cells into neurons (Francis et al, Neuron 23(2):247-56, 1999). Our results show decreased expression of doublecortin during differentiation of neural stem cells into astrocytes, suggesting the absence of neurogenesis in our differentiation protocol. We have clarified this in the results section on page 10.

  1. Please use official gene names and put them above graphs when the authors show results of mRNA expression levels.

Response: As suggested by the reviewer, we have added the corresponding gene names above the graphs in Fig. 2 and Fig. 4.

  1. In Fig. 4D and E, the labeling of X-axis and Y-axis must be incorrect, respectively.

Response: We apologize for this oversight and thank the reviewer for bringing this to our notice. We have made the necessary changes in Fig. 4 D and E.

  1. Please explain the rationale for choosing 48 h of hypoxia exposure in more detail.

Response: The hypoxia exposure time was determined primarily based on our initial preliminary studies and published reports (Zhou et al, Front Physiol 12:729925, 2021; Engelhardt et al, Fluids Barriers CNS 12:4, 2015).  Our preliminary results showed that a minimum of 48 hours of exposure to hypoxic conditions is essential for consistent upregulation of the hypoxia markers CA-9 and HIF-1α in cultures of differentiated astrocytes. We have provided the rationale for the duration of hypoxia exposure in the revised manuscript in the discussion section, on page 16.  

  1. The authors are recommended to examine the effects of hypoxia on EAAT1 protein levels in astrocytes.

Response:  We appreciate the suggestion by the reviewer. However, although we examined the expression of both EAAT1 and EAAT2 mRNAs for characterization of astrocytes differentiated from neural stem cells, the present study is focused on the primary glutamate transporter EAAT2, which is known to be responsible for up to 90% of the glutamate uptake in the forebrain and hippocampus (Robinson, Neurochem Int, 33(6):479-91, 1999; Zhou et al, J Neurosci, 34(40):13472-85, 2014), and we have determined both the RNA and protein levels of EAAT2 under different experimental conditions.  

  1. It has been demonstrated that hypoxia induces the release of interleukin-1beta (IL-1b) from astrocytes via HIF-1 (Zhang et al., J Neuroimmunol, 2006). In addition, it has been suggested that hypoxia and IL-1b cooperatively induce the release of glutamate from astrocytes (Fogal el al., J Neurosci, 2007; Vangeison and Rempe, Neuroscientist, 2009). Although glutamate uptake has been assessed using radiolabeled glutamate in many previous reports, the authors assessed glutamate uptake by measuring glutamate concentrations in culture supernatant. Therefore, it is possible that glutamate concentrations in HBSS were affected not only by glutamate uptake but also by glutamate released by astrocytes in the culture and assay system of this report. The authors are recommended to confirm if glutamate release from astrocytes is stimulated in their culture system.

Response: We thank the reviewer for the constructive suggestion. However, our results (controls) do not suggest substantial release of glutamate by astrocytes into the culture media. In our experiments, after exposing astrocytes to hypoxia for 48 hours, the culture media were replaced with HBSS containing 2 mM glutamate. After a 30 min incubation, glutamate concentrations were assayed in the supernatant to determine glutamate uptake by the astrocytes. We found that the glutamate concentrations in the supernatant decreased after the 30 min incubation from 2.3 ± 0.5 mM in control, to approximately 1.6-1.7 mM in all treatment groups, including normoxic controls (Results page 11 and supplementary figure S1) signifying that glutamate was predominantly being taken up by the cells after exposing them to hypoxia. However, with reference to the study by Fogal et al (2007) that used a 24-hour incubation with IL-1b, we measured glutamate concentrations only after 30 minutes of incubation, which is potentially a short period of time for a significant accumulation of inflammatory mediators and stimulation of glutamate release from astrocytes. We have cited this work by Fogal et al and discussed it in the revised manuscript in the discussion section, on page 15.

  1. The authors cited a report by Santello et al. (ref. 34) for Glutamate uptake assay. However, this is a review article and there is no information about the procedure of glutamate uptake assay in the report. Please cite an appropriate report.

Response:  We apologize for this error. The appropriate reference has been added in the revised manuscript.

  1. Not all product number information is provided.

Response:  We checked the manuscript and have included all the missing product information.

References:

  • Engelhardt S, Huang SF, Patkar S, Gassmann M, Ogunshola OO. Differential responses of blood-brain barrier associated cells to hypoxia and ischemia: a comparative study. Fluids Barriers CNS. 2015 Feb 17;12:4. doi: 10.1186/2045-8118-12-4.
  • Francis F, Koulakoff A, Boucher D, Chafey P, Schaar B, Vinet MC, Friocourt G, McDonnell N, Reiner O, Kahn A, McConnell SK, Berwald-Netter Y, Denoulet P, Chelly J. Doublecortin is a developmentally regulated, microtubule-associated protein expressed in migrating and differentiating neurons. Neuron. 1999 Jun;23(2):247-56. doi: 10.1016/s0896-6273(00)80777-1.
  • Robinson MB. The family of sodium-dependent glutamate transporters: a focus on the GLT-1/EAAT2 subtype. Neurochem Int. 1998 Dec;33(6):479-91. doi: 10.1016/s0197-0186(98)00055-2.
  • Zhou S, Zhong Z, Huang P, Xiang B, Li X, Dong H, Zhang G, Wu Y, Li P. IL-6/STAT3 Induced Neuron Apoptosis in Hypoxia by Downregulating ATF6 Expression. Front Physiol. 2021 Oct 21;12:729925. doi: 10.3389/fphys.2021.729925.
  • Zhou Y, Wang X, Tzingounis AV, Danbolt NC, Larsson HP. EAAT2 (GLT-1; slc1a2) glutamate transporters reconstituted in liposomes argues against heteroexchange being substantially faster than net uptake. J Neurosci. 2014 Oct 1;34(40):13472-85. doi: 10.1523/JNEUROSCI.2282-14.2014.

Reviewer 2 Report

The study is appropriately conceived, executed and well written. However, I have an issue with the monoculture model demonstrating glutamate uptake. The authors should address these concerns or add them to the work limitations.

  1. There is a prominent role for contact-dependent neuron-to-astrocyte and/or endothelial cell-to-astrocyte Notch signaling for inducing and maintaining the expression of these astrocytic glutamate transporters. However, in the study design, the inducing signal cannot be elicited. How do you compensate for this?

  1. Neural co-cultures can better mimic the complex interactions between cells and provide a more robust platform for studying neurodevelopment; because glutaminergic neurons are completely missing from your culture. Therefore, you won't expect the physiological level of glutamate and consequently a suboptimal glutamate clearance by the astrocytes. How do you explain this limitation in your study?

Author Response

Response to Reviewer's Comments

Manuscript ID: genes-1598699

Title: Glutamate uptake is not impaired by hypoxia in a culture model of human fetal neural stem cell-derived astrocytes.

We are thankful to the reviewers for critically reviewing the manuscript and giving their valuable suggestions for its improvement.  We are gratified that the reviewers find our work of importance to the field. All suggestions have been incorporated into the revised manuscript. We have addressed all points, including comments, questions, and suggestions given by reviewers and marked the major changes in the manuscript.

Reviewer #2:

Comments: The study is appropriately conceived, executed and well written. However, I have an issue with the monoculture model demonstrating glutamate uptake. The authors should address these concerns or add them to the work limitations.

  1. There is a prominent role for contact-dependent neuron-to-astrocyte and/or endothelial cell-to-astrocyte Notch signaling for inducing and maintaining the expression of these astrocytic glutamate transporters. However, in the study design, the inducing signal cannot be elicited. How do you compensate for this?
  2. Neural co-cultures can better mimic the complex interactions between cells and provide a more robust platform for studying neurodevelopment; because glutaminergic neurons are completely missing from your culture. Therefore, you won't expect the physiological level of glutamate and consequently a suboptimal glutamate clearance by the astrocytes. How do you explain this limitation in your study?

Response: We thank the reviewer for appreciation of our work. We agree with the reviewer that in vitro studies are limited as they cannot model the dynamic in vivo environment where numerous pathways and cell types are in constant communication within a complex organ. However, in vitro and in vivo studies each have advantages and disadvantages, and the main goal of this study was to develop a novel culture model of human fetal neural stem cell (FNSC)-derived astrocytes and examine their glutamate uptake in response to hypoxia, as the roles and contributions of various human fetal neural cells in the development of glutamate excitotoxicity are not fully understood. We successfully developed and characterized cultures of FNSC-derived astrocytes from aborted human fetal brains and demonstrated that these astrocytes express glutamate transporters and can maintain glutamate uptake under hypoxic conditions. However, we agree with the reviewer that this study merits further investigation using in vivo and complex coculture systems of glial and neuronal cells in the development of glutamate excitotoxicity and in hypoxic ischemic brain injuries. We have incorporated the limitations of our study (page 16) in the revised manuscript as suggested by the reviewer.

Round 2

Reviewer 1 Report

The authors responded to each query of the reviewer satisfactorily. The only thing needed to be done is that gene symbols should be italicized.